# Redox Signaling Modulates Activity of Immune Checkpoint Inhibitors in Cancer Patients

**DOI:** 10.3390/biomedicines11051325

**Published:** 2023-04-29

**Authors:** Alessandro Allegra, Giuseppe Murdaca, Giuseppe Mirabile, Sebastiano Gangemi

**Affiliations:** 1Division of Hematology, Department of Human Pathology in Adulthood and Childhood “Gaetano Barresi”, University of Messina, 98125 Messina, Italy; aallegra@unime.it; 2Department of Internal Medicine, Ospedale Policlinico San Martino IRCCS, University of Genova, Viale Benedetto XV, n. 6, 16132 Genova, Italy; 3Allergy and Clinical Immunology Unit, Department of Clinical and Experimental Medicine, University of Messina, 98125 Messina, Italy; gangemis@unime.it

**Keywords:** immune checkpoint, immune check points inhibitor, oxidative stress, cancer, immune system, immunotherapy, immunosurveillance, T lymphocyte, PD-1, PD-L1

## Abstract

Although immunotherapy is already a staple of cancer care, many patients may not benefit from these cutting-edge treatments. A crucial field of research now focuses on figuring out how to improve treatment efficacy and assess the resistance mechanisms underlying this uneven response. For a good response, immune-based treatments, in particular immune checkpoint inhibitors, rely on a strong infiltration of T cells into the tumour microenvironment. The severe metabolic environment that immune cells must endure can drastically reduce effector activity. These immune dysregulation-related tumour-mediated perturbations include oxidative stress, which can encourage lipid peroxidation, ER stress, and T regulatory cells dysfunction. In this review, we have made an effort to characterize the status of immunological checkpoints, the degree of oxidative stress, and the part that latter plays in determining the therapeutic impact of immunological check point inhibitors in different neoplastic diseases. In the second section of the review, we will make an effort to assess new therapeutic possibilities that, by affecting redox signalling, may modify the effectiveness of immunological treatment.

## 1. Introduction

### 1.1. General Considerations on Immunological Checkpoints and Oxidative Stress in Neoplasms

One strategy tumour cells have developed to evade immune surveillance is a changed expression of the immunological checkpoint receptor programmed death receptor 1 (PD-1) and its ligand, PD-L1 [1]. The ligand–receptor interactions that lead to the activation of numerous immunological checkpoints also involve the cytotoxic T lymphocyte-associated protein-4 (CTLA4), PDL2 receptors expressed on immune and tumour cells, and others [2] (Figure 1). When PD-1 on cytotoxic T-cells binds to PD-L1 on the surface of cancer cells, this inhibits T lymphocyte stimulation and immune escape. Probably, the primary method of immunological resistance to cancer is the activation of specific immune checkpoint mechanisms.

Immunotherapy reduces immunological tolerance by preventing the communication between tumour cells and the immune system. Hence, it is anticipated that blocking immunological checkpoints will be a new cancer therapeutic method [1]. In fact, to prevent the contact and hence restore antitumour immunity, monoclonal antibodies (mAbs) targeting PD-1 or PDL1 have been produced. Over the past few years, a number of mAbs have been registered in various tumour pathologies, and a number of further anti-PD-(L)1 mAbs are presently undergoing clinical progress. The similar goal of blocking the checkpoint and activating T cell-based immunotreatment has also been achieved via the development of peptides and small compounds that target PD-L1 [3]. Increasing evidence demonstrates that immunotherapy techniques are highly effective at eliminating tumours, preventing their reappearance, and have potential for future application [4].

Immune checkpoint inhibitor (ICI) medications have had significant success; however, most patients who receive ICI monotherapy do not experience sufficient long-term anticancer effects [5]. The so-called “cold tumour” caused by defects in antigen presentation to T cells, absence of T cell activation, lack or minority of activated T cell infiltration in tumour tissues, and abundance of immune suppressor cells such as regulatory T cells (Tregs) and myeloid-derived suppressor cells (MDSCs) is one of the main indicators of a poor response to ICI therapy [6,7,8,9].

As a result, recent research on immunogenic cell death (ICD) has been conducted extensively to study the field of cancer immunotherapy [10,11] and ways of reducing oxidative stress (OS), which led to an improved immune response against the tumour [12]. In fact, OS is one of the most representative biological situations that cancer cells are typically exposed to [13,14]. The raised intracellular amounts of reactive oxygen species (ROS) are a feature of the cancer milieu, which is characterized by increased OS [15]. ROS are extremely reactive oxygen compounds that include hydrogen peroxide, superoxide, peroxides, and hydroxyl radicals. The activation of specific oncogenes, hypoxia, and external stimuli such as chemotherapy and radiotherapy can all be linked to dysregulated ROS in tumours cells.

Owed to abnormalities in DNA repair, protein degradation, and lipid peroxidation, excessive ROS production might be fatal to cancer cells [16,17,18]. Consequently, OS has a significant function in the emergence and growth of cancers as well as the management of neoplastic illnesses [19]. The intracellular biological redox steady state is disrupted when cells are continuously exposed to environmental stress, such as UV radiation, metabolic stress, and anti-cancer medications. Excessive ROS are then produced, which affects immune dysfunction, signal transduction, cell growth, and cell death [20]. However, as mentioned above, ROS may also cause DNA base modifications or sequence rearrangements, DNA damage-derived miscoding lesions, and oncogene activation, all of which work in concert to promote the growth and spread of tumours [21] (Figure 2). In actuality, OS has a role in a number of malignancies, including those related to the brain, breast, pancreatic adenocarcinoma, lung cancer, and others [22,23,24,25,26,27,28].

### 1.2. Oxidative Stress and Immune System

Both immune effectors and tumour cells are influenced in a tumour microenvironment (TME) when ROS concentrations are maintained at elevated levels. According to certain studies, immune cells’ ability to act as antioxidants plays a role in their ability to combat cancer [29,30]. Immune suppression takes place in the TME when the ROS level rises to prevent immune cells from destroying tumours [31]. Furthermore, knowledge of the effects of ROS on dendritic cells (DC), macrophages, natural killer (NK) cells, T cells, and B cells has increased [32,33,34]. Because it affects both tumour cells and TME, oxidative stress has been shown to either promote or inhibit the growth or spread of tumours [35].

Moreover, ROS have a substantial impact on how PD-1 and PD-L1 are expressed, although it is not always clear how ROS and PD-(L)1 interact. Nonetheless, a number of studies have indicated that ROS regulate PD-L1 expression [36]. Particularly important, a considerable increase in the expression of PD-L1 was observed when cells were exposed to a number of ROS inducers, including buthionine sulphoximine and the anticancer drug paclitaxel. [37]. Many natural items, well-known medications, and experimental chemicals were found to modulate the expression of PD-L1 and ROS.

An important component of redox control and antioxidant defence is the thioredoxin (Trx) system. It is made up of the crucial anticancer targets Trx and thioredoxin reductase (TrxR). Anticancer drugs frequently target the Trx/TrxR system. The thi-oredoxin reductase 1 (TrxR1) enzyme, for instance, is a vital intracellular redox sensor and antioxidant enzyme that is frequently overexpressed in a range of cancer types. It is inhibited by the organoselenium chemical ethaselen (BBSKE). Through the formation of two covalent connections with the cysteine-497 and selenocystine-498 residues, BBSKE particularly targets the C-terminal redox core of TrxR [38]. BBSKE’s suppression of TrxR1 causes cells to produce more ROS due to inhibition of scavenging [39]. When taken with other anticancer drugs such sodium selenite [40] and the multi-targeted tyrosine kinase inhibitor sunitinib [39], BBSKE has demonstrated a synergistic effect. According to a recent publication, TrxR1 is inhibited by BBSKE, or the enzyme is knocked down, in cancer cells, which lowers the amount of PD-L1 expression [41]. Nevertheless, prior to BBSKE exposure, a therapy with the antioxidant N-acetylcysteine restored PD-L1 expression.

Similarly, the antipsychotic medicine trifluoperazine (TFP), which is also taken into consideration for the treatment of cancer [42,43], deserves to be included as another ROS-inducing substance. Recent research has shown that TFP increases ROS levels in colorectal cancer cells while also raising PD-L1 expression in these cancer cells and PD-1 expression in CD4+ and CD8+ T cells that infiltrate tumours [42].

Metformin and phenformin, two biguanide medications used to treat diabetes, have also demonstrated anticancer activity both in vitro and in vivo. They both promote oxidative stress-mediated apoptosis in cancer cells and inhibit PD-L1 expression, mainly via the Hippo signalling pathway [44,45,46,47]. They also both enhance the production of ROS. It has been shown that metformin promotes the interaction and phosphorylation of PD-L1 by the AMP-activated protein kinase (AMPK) protein, leading to its abnormal glycosylation and subsequent destruction [48,49]. They appear to increase the anticancer activity of PD-1 inhibition, not diminish it, and hence they do not reduce the efficacy of anti-PD-1 therapy [50]. Immune checkpoint inhibitors combined with metformin were given to subjects affected by melanoma or lung cancer [51,52]. By suppressing myeloid-derived suppressor cells and lowering tumour cell oxygen consumption, these drugs might amplify PD-1 blocking. Additionally, this would diminish intratumoural hypoxia [53].

The catechin by-product EGCG (epigallocatechin-3-gallate), which is prevalent in green tea, can likewise lessen intracellular ROS production and stop the loss of antioxidants. This natural substance can reduce the OS brought on by several stimuli, including, for instance, arsenic and cigarette smoke [54,55,56]. It is a powerful immune–epigenetic modulator for the treatment and/or prevention of cancer [57,58] and has a variety of targets, comprising histone deacetylases and metalloproteinases. Its anti-oxidative and free radical scavenging properties have received a lot of attention [59]. It is interesting to note that PD-L1 expression was shown to be decreased by EGCG in pulmonary tumour cell lines, and that PD-L1 suppression by EGCG led to a recovery of T cell function [60].

The action of aryl hydrocarbon receptor (AhR) appears to be more complex [61]. Initially, AhR was identified as a transcription factor controlling xenobiotic response [62]. As a result of ligand interaction, AhR is translocated into the nucleus where it heterodimerizes with Aryl hydrocarbon Receptor Nuclear Translocator (ARNT) to create an active transcription complex. In the absence of a ligand, AhR aggregates in the cytoplasm with other chaperone proteins. The cytochrome P450 enzymes, which include CYP1A1, CYP1A2, and CYP1B1, are a subfamily of metabolizing enzymes that are recognized by the AhR/ARNT complex as xenobiotic-responsive elements (XREs) [63]. In addition to controlling immunological checkpoint protein expression, AhR is essential in modulating immune tolerance and immune suppression [64]. For instance, AhR is involved in PD-1 and PD-L1 transcriptional activation [65,66]. Another enzyme implicated in immunological suppression, indoleamine 2′ 3′-dioxygenase 1 (IDO1), is transactivated by AhR as well [67].

Tryptophan (Trp) is an amino acid that IDO1 uses to enzymatically convert into kynurenine (Kyn), an oncometabolite that can decrease T cell cytotoxicity in the tumour microenvironment by depleting Trp [68]. A study revealed that the hydrogen peroxide-treated human keratinocyte cell line (HaCaT) had increased AhR activity, resulting in increased expression of its downstream targets, including cytochrome P450 genes. Intriguingly, AhR activation and its downstream signalling were increased by preincubating the whole culture media with hydrogen peroxide. The oxidant causes the synthesis of oxindole, a tryptophan catabolic product, according to a later mass spectrometric investigation. The fact that 2-oxindole can activate AhR was also demonstrated by the authors, strongly indicating that ROS may have a considerable influence on AhR signalling [69,70]. AhR’s role as an oncogene is mostly explained by the ROS accumulation caused by enhanced CYP activities, which favours malignant transformation by causing severe OS and increased DNA damage [71]. High AhR expression in breast cancer cells is significantly correlated with ROS build-up, which causes AhR to translocate into the nucleus and enhances its transcriptional activity [72,73].

In any case, although immune checkpoint inhibitors have transformed the way that cancer is treated, only a small percentage of patients get long-lasting improvements. Therefore, it is crucial to comprehend the connections between ROS and the PD-(L)1 checkpoint, particularly to aid in the development of novel medication combinations. Yet, as demonstrated above, the relationship between PD-L1 presence and ROS generation is complicated. ROS have the ability to either increase or decrease the amount of PD-L1 in cancer cells. PD-L1 expression is frequently encouraged by increased ROS production, and vice versa, ROS scavenging can inhibit PD-L1. In spite of that, there are clear exceptions when it comes to medications that boost ROS production while lowering PD-L1 expression and vice versa. Although no clear and distinct association can be inferred, drugs that alter ROS generation can have a considerable impact on PD-L1 expression [74] (Figure 3).

For instance, Auronafin and disulfiram are two ROS-producing medications that have been shown in studies to increase PD-L1 expression in cancer cells, whereas other ROS-enhancing compounds (such as ethaselen, chaetocin, and metformin) tend to lower PD-L1 expression. PD-L1 expression is increased in cells treated with As_2_O_3_ and disulfiram, but ROS scavenging is shown simultaneously with PD-L1 downregulation. However, in many instances (after cell treatment with the plant extract Anoectochilus formosanus), ROS production and PD-L1 expression showed a similar pattern. It has been discovered that human oncoviruses such as the EBV, which infect human primary monocytes to significantly increase the expression of PD-L1 on their surface by producing ROS, follow a similar pattern [75].

The impact of ROS-inducing drugs on the expression of PD-L1 is undoubtedly more complicated than has previously been documented, and these chemicals may also have additional modes of action. For instance, an alternative mechanism used by AhR could be the excess kynurenine produced by malignant cells and the impact of kynurenine on TME immune cells. Kynurenine may increase the expression of PD-L1 on G-MDSC, macrophages, and DCs, all of which express AhR, just as it does in cancerous cells. Through direct transcriptional regulation, AhR activation also produces the immunosuppressive CD39 ectoenzyme on macrophages and T cells. Furthermore, melanomas’ production of AhR and “transcellular” Kynurenine has been connected to PD-1 expression on CD8+ T cells in the TME. Chronic IFN production appears likely to aggravate each of these AhR-mediated consequences. It was surprising to learn that a large portion of the effect of IFN on Cd274 transcription in the MOC1 model is due to the AhR. It was also remarkable that the recognized IFN-induced induction of Ido was mostly AhR controlled. Although there have been suggestions that IFN and AhR signalling interact, it has been reported that AhR control of IFN-driven outcomes, notably PD-L1 and IDO activation, can be demonstrated in the context of cancer [65].

The condition of the check points and the redox balance in distinct neoplasms will be discussed in more detail in the sections that follow, as well as how changing OS might affect how well check point inhibitors prevent tumour growth in particular tumour pathologies.

## 2. Cancer, Oxidative Stress, and Immune Check Point Inhibitors

### 2.1. Breast Cancer

With no effective targeted therapy, triple-negative breast cancer (TNBCs) is a particularly deadly and aggressive kind of breast cancer. Just 25–30% of TNBC patients respond to neoadjuvant chemotherapy or radiotherapy, which continues to be the cornerstone of treatment. Thus, there is an unmet clinical need to create new TNBC therapy approaches. However, although anti-PD-L1 monoclonal antibodies have demonstrated a high clinical efficacy in other tumours, TNBCs have not responded well to anti-PDL1 monotherapy. In the Keynote 012 study, only 18.5% of TNBC patients showed response to pembrolizumab monotherapy [76]. Similarly, 8.6% of patients with locally advanced or metastatic TNBC and 26% of those with metastatic TNBC responded to avelumab and atezolizumab, respectively [77], and it has been suggested that in TNBCs, anti-PD-L1 monotherapy may not be as effective as a combination of PD-L1-targeting treatments and other targeted medicines [78].

The increased intracellular OS that leads to BC is a major factor in its pathogenesis [79]. Antioxidants that are upregulated, such as Trx and glutathione (GSH), shield cancer cells from this heightened OS and give them a survival advantage [80,81,82]. Glutathione reductase (GSR) and TrxR1 are redox enzymes that maintain the reduced forms of GSH and Trx, hence maintaining the efficiency of both antioxidant systems. When there is no antioxidant activity, these two systems functionally balance one another out. When the GSH system in lung cancer cells is downregulated, the Trx system is then upregulated, making the cancer cells functionally reliant on Trx. TNBC cells are therefore more vulnerable to higher levels of oxidative stress despite having increased intracellular OS and decreased glutathione.

As for BC, the expression of the Trx pathway genes is significantly elevated in TNBC patients compared to non-TNBC patients and is connected with poor survival outcomes, according to a study that evaluated a panel of antioxidant genes using The Cancer Genome Atlas (TCGA) and METABRIC databases [83]. Hence, a number of compounds have been used to reduce OS, including Auranofin (AF), a gold (I)-containing molecule that was authorized in 1985 as a main treatment for rheumatoid arthritis [84]. By interfering with the redox system inside the cell, AF has been shown to behave as a pro-oxidant agent as one of its main modes of action. This is accomplished by two selenoenzyme isoforms inhibiting TrxRs. TrxRs bind to around four triethylphosphinenegold (I) cations (AuPet3+) fragments, and biochemical experiments show that the gold compound dramatically alters the active selenocysteine site of the enzymes. TrxRs also have a selenocysteine moiety that is redox active. It seems that the cytotoxicity of AF is influenced by the inhibition of both mitochondrial TrxR2 and cytoplasmic TrxR1. TrxRs are NADPH-dependent because they transfer electrons from NADPH to the active disulfide site on the oxidized Trx protein in order to enable Trx to function. In this way, Trx catalyses the reduction of ROS from oxidized cysteines of proteins and in the process, Trx itself becomes oxidized [85]. The interaction between the active site dithiol in reduced Trx and oxidized cysteines of many proteins induces the process of thiol/disulfide exchange reaction to form an oxidized Trx.

Treatment with AF reduced the development of TNBC cells produced as spheroids and caused selective cell death. Moreover, AF therapy significantly reduced Trx redox activity in a number of TNBC models, including the syngeneic 4T1.2 model, MDA-MB-231 xenograft, and patient-derived tumour xenograft. For the first time, the investigation demonstrated that AF boosted CD8+Ve T-cell tumour infiltration in vivo and modified immunological checkpoint PD-L1 expression in an ERK1/2-MYC-dependent way. Moreover, AF and antiPD-L1 antibody together effectively slowed the growth of 4T1.2 primary tumours. These studies offer a potential therapeutic approach that could be used in TNBC patients [83] that combines AF with an anti-PD-L1 antibody.

### 2.2. Ovarian Cancer

The most common pathological subtype of ovarian cancer (OC), which kills women in developed nations, is epithelial ovarian cancer (EOC) [86,87]. Advanced epithelial ovarian cancer has an extremely high probability of recurrence, and its five-year survival rate is just about 30%. Additionally, the effectiveness of treatment for individuals with recurrent ovarian cancer frequently falls short of the initial therapy, and the length of time it takes to establish remission following therapy shortens with each additional recurrence.

On OC, OS has a variety of intricate impacts. The imbalance of antioxidant processes causes OS levels to typically be significantly greater in OC patients, according to pertinent findings [88,89,90]. According to a study, The hydroxyl radical created by the Fenton reaction can cause DNA double-strand breaks (DSB) in the fallopian tubal epithelium, which can hasten the progression of OC [91]. More importantly, it has been shown that a number of redox-modified signalling pathways, such as the Wnt/-catenin signalling network, the AKT/mTOR signalling pathway, the Nrf2/PGC1 signalling pathway, and the Notch signalling system [92,93,94,95], are essential in the pathogenesis of OC.

For example, the Wnt/-catenin signalling pathway can be activated by the nucleoredoxin oxidized by ROS [96], and the active Wnt/-catenin signalling route has been linked to the increased platinum resistance in OC [97]. By influencing immune cells and metabolites in the tumour microenvironment (TME), OS also contributes to the development of OC [98,99]. Advanced OC patients’ neutrophils have amplified functional activities compared to healthy women, and stimuli dramatically enhance the amount of ROS that are produced [100].

OS has some bearing on the treatment outcome in OC patients, as demonstrated by the earlier investigations. OS can specifically influence chemoresistance by causing point mutations in important redox enzymes [101]. Moreover, immune cells produce ROS, one of their key second messengers, which opens the door for the use of antioxidants in immunomodulatory therapy [102].

Oxidative stress-correlated genes (OSRGs) were sourced from the Molecular Signatures Database for one investigation. Moreover, The TCGA gene expression profiles and clinical data were used to find the prognostic OSRGs. Furthermore, successive analyses were performed to create a predictive signature that was later on verified indifferent Gene Expression Omnibus (GEO) datasets. The role of the OSRG signature in immunotherapy was then further investigated using immunological checkpoint genes (ICGs), the tumour immune dysfunction and exclusion (TIDE) algorithm, the GSE78220 and IMvigor210 and cohorts. The importance of the OSRG signature in chemotherapy was further examined using the Genomics of Drug Sensitivity in Cancer (GDSC) and CellMiner databases. At the conclusion of the investigation, 34 prognostic OSRGs were found, and 14 of them were selected to create the most effective prognostic signature. The prognosis of patients with lower OS-related risk scores was better, and OC subjects in the low-risk category may have responded more favourably to immunotherapy. Furthermore, results showed that anti-PD-1/L1 immunotherapy was more likely to be beneficial for patients with lower OS-related risk scores. These findings suggest that the OSRG profile might serve as a potent prognostic factor for OC, which helps develop more specialized treatment plans for OC subjects. Immunotherapy affects immunological subgroups differently, and as a result, diverse clinical developments may result [103].

Based on possible signalling pathway and TMB level connections with immunotherapy, a study considered the immunotherapeutic responses of patients in different risk groups. Immune dysfunction was calculated for each patient using the TIDE module in order to predict the immunotherapeutic response. Patients in the low-risk group had a higher chance of responding favourably to immunotherapy, but those in the high-risk group were more susceptible to immunological escape and dysfunction. Last but not least, it was determined that the OS-related risk score could predict the clinical response to anti-PD-1/L1 immunotherapy using the IMvigor210 and GSE78220 datasets. This finding showed that patients in the high-risk group with lower OS-related risk scores may be more likely to benefit from anti-PD-1/L1 immunotherapy because they had significantly lower responses than those in the low-risk group and because responders in both datasets had OS-related risk scores that were noticeably lower than those of non-responders [104].

### 2.3. Endometrial Cancer

The incidence of endometrial cancer (EC), which is second only to cervical cancer in terms of frequency of diagnosis, is among the most common gynaecological cancers. Every year, EC claims the lives of more than 50,000 women globally [105,106]. Due to the missed opportunity for surgery and efficient chemotherapy treatments for endometrial cancer patients with metastases, the prognosis for EC is still dismal. Therefore, it is essential to investigate efficient therapeutic options for endometrial cancer.

Targeted therapy and immunotherapy have recently been shown to be more dependable and effective EC treatment options [107,108]. Moreover, several research have examined the links between immunotherapy and the modifications of the redox system in patients with EC and how these variations may affect the effectiveness of immune therapy.

The solute carrier family 7 member 11 (SLC7A11) is widely overexpressed in cancers and is well-known for its role in maintaining intracellular glutathione levels and preventing oxidative stress-induced cell death, such as ferroptosis [109]. It is the primary regulator of this type of apoptosis, and ferroptosis is marked by a reduction in its expression level [110,111,112]. In a study, researchers visualized the expression of SLC7A11 across EC, assessed its prognostic significance, and examined its relationships with immune cell infiltrates and immunological biomarkers. Additionally, they used data mining to thoroughly investigate whether SLC7A11 overexpression is connected to how well immunotherapy works for cancer patients. In individuals with uterine corpus endometrial cancer (UCEC), SLC7A11 expression was markedly increased and correlated with prognosis. In comparison to individuals with UCEC, normal participants had considerably greater DNA methylation levels in the SLC7A11-promoter region. Additionally, they showed a relationship between SLC7A11 overexpression and the immune checkpoint blocker (ICB), tumour immune induction complex (TIIC), and immunotherapy responsiveness. As a result, SLC7A11 overexpression is linked to both the effectiveness of immunotherapy and a good prognosis for patients with UCEC [113].

### 2.4. Melanoma

The skin cancer type that is most deadly, cutaneous melanoma, is challenging to cure and has gained attention recently due to an increase in incidence around the world. Resistance, serious side effects, and a poor quality of life have all been connected to the use of antitumoral therapies for this neoplasm [114]. Cancer immunotherapies are now often utilized as adjuvant and neoadjuvant therapy in melanoma patients and have improved previously low survival rates, such as checkpoint inhibitors ipilimumab, nivolumab, and pembrolizumab [115,116].

As for the OS in this disease, the pro-survival cellular responses to OS rely heavily on the KEAP1-NRF2 pathway [117,118,119]. The E3 ubiquitin ligase adaptor KEAP1 negatively regulates the transcription factor NRF2 by targeting it for ubiquitination and proteasome-dependent degradation, which keeps NRF2 at low levels under basal, non-stressed conditions [120,121,122]. Several cysteine-based stress sensors found in the KEAP1 protein allow it to modulate its function in response to changes in the cellular redox state [123,124,125,126].

In addition to traditional chemotherapeutics, NRF2 hyperactivation can make tumours resistant to immune checkpoint inhibitor therapies, and it is suggested with examples that using a synthetic lethal strategy mediated by NRF2-target gene-dependent bioactivation of prodrugs represents a promising strategy to specifically enhance toxicity to previously incurable NRF2-hyperactivated human tumours [127].

Moreover, methylglyoxal (MG), an oncometabolite implicated in metabolic reprogramming, is detoxified by glyoxalase 1 (encoded by GLO1), a glutathione-dependent enzyme. GLO1 has recently been shown to be overexpressed in human malignant melanoma cells and patient tumours, supporting its new function as a molecular regulator of invasion and metastasis in melanoma. It was recently revealed that CRISPR/Cas 9-based GLO1 deletion from human A375 malignant melanoma cells changes redox homeostasis, which is associated with acceleration of carcinogenesis. This was carried out using NanoStringTM gene expression profiling. TXNIP, a master regulator of cellular energy metabolism and redox homeostasis, was found via NanostringTM analysis to exhibit the most dramatic expression shift in response to GLO1 removal. The use of the pharmacological GLO1 inhibitor TLSC702 matched the effects of GLO1 KO, indicating that GLO1 regulates MG to control the expression of TXNIP. Reduced glucose uptake and metabolism with downregulation of gene expression (GLUT1, GFAT1, GFAT2, LDHA), depletion of related critical metabolites (glucose-6-phosphate, UDP-N-acetylglucosamine), and immune checkpoint modification were characteristics of the GLO1 KO condition (PDL1).

The authors noted that GLO1 KO melanoma cells displayed a shortened population doubling time, cell cycle alteration with increased M-phase population, and enhanced anchorage-independent growth, a phenotype supported by expression analysis, while confirming an earlier finding that GLO1 deletion limits invasion and metastasis with modulation of EMTrelated genes (e.g., TGFBI, MMP9, ANGPTL4, TLR4, SERPINF1) (CXCL8, CD24, IL1A, CDKN1A). A375 melanoma tumour development and metastasis can be dysregulated in opposite ways as a result of GLO1 deletion, as was shown by the observation of an enhanced growth rate of GLO1 KO tumours in a SCID mouse melanoma xenograft model, together with TXNIP overexpression and metabolic reprogramming [128].

Lastly, several studies focused on the expression of PDL1 expressed by melanoma cells [129,130,131,132,133] because it has recently been demonstrated that tumour glucose metabolism (as associated with high flux through glycolysis and the hexosamine pathway) modulates expression of specific genes relevant to cancer cell immune evasion. Moreover, the study found that GLO1 KO condition reduced PDL1 expression, both at the mRNA and protein levels (GLO1 WT versus GLO1 KO [B40 and C2]), a noteworthy finding given the significance of PD-L1 as a key target for clinically relevant melanoma immunotherapeutic intervention [128].

### 2.5. Glioma and Glioblastoma Multiform

A central nervous system tumour known as a glioma can affect the brain and spinal cord. It makes up 81% of craniocerebral malignancies and is the major tumour of the central nervous system that occurs the most frequently. Glioma is divided into four main forms by the World Health Organization, with grade IV glioblastoma multiform (GBM) being the most prevalent and aggressive variety.

Adenosine triphosphate (ATP) is produced by normal cells’ mitochondria through oxidative phosphorylation (OXPHOS). Adenosine diphosphate (ADP) would be converted into ATP on the mitochondrial membrane via the chemical gradient-driven electron transport chain during the OXPHOS [129,130,131,132,133,134,135]. The Warburg effect states that healthy cells rely on OXPHOS for energy while malignant cells use aerobic glycolysis as their primary energy source [136,137,138].

Several studies have demonstrated that cancer cells’ mitochondrial architecture and functions are aberrant, and that these redox abnormalities may encourage cell growth and even metastasis. In a study, the cell viability of the U87 MG, GBM cell linesT98G, GBM 8401, and U138 MG was assessed using isoaaptamine and aaptamine. The results demonstrated that in these four cell lines, isoaaptamine was more effective than its iso-form, aaptamine, and that GBM 8401 was the most responsive to isoaaptamine. The research on GBM 8401 cells demonstrated that isoaaptamine caused an increase in cleaved caspase 3 and poly ADP-ribose polymerase (PARP), which led to the induction of apoptosis. Moreover, isoaaptamine increased the amounts of ROS, inhibited SOD1 and SOD2 in mitochondria and cells, and altered the potential of the mitochondrial membrane. The oxygen consumption rates and activity of mitochondrial complexes I through V also significantly decreased. After being treated with isoaaptamine, mitochondrial dynamics tended to fission rather than fusion, and ATP production was eliminated. Furthermore, autophagy was activated as evidenced by the formation of acidic organelle vesicles [139].

Several other research have discovered a link between changes in the immune system, immunological check points, and the redox system. One such example is the ability of tumour-infiltrating immune cells (TIICs) in the TME to quicken the proliferation of tumour cells [140,141]. The likelihood that tumour-associated macrophages (TAMs) encourage glioma cell growth and invasion is growing [142]. Interestingly, TAM suppression effectively prevents the development of gliomas [143].

Furthermore, disorders of the structural and functional neural systems may result from oxidative stress, which plays a role in central nervous system illnesses. Overproduction of ROS and reactive nitrogen species (RNS) during oxidative stress can result in neuronal malfunction and death [144]. KLHDC8A was identified as a hub gene via differential expression analysis using bioinformatics methods. Proteins from the Kelch superfamily that include Kelch domains are encoded by KLHDC8A [145]. Many human diseases, including cancer and neurological disorders, depend on the function of several Kelch proteins. In cancer, Kelch protein expression is increased [146,147]. The proliferation, migration, and invasion of glioma cells are all triggered by the overexpression of KLHDC8A [148].

A study showed that KLHDC8A was expressed by both tumour cells and macrophages associated with tumours. When compared to healthy brain tissues, glioma tissues had higher levels of KLHDC8A expression, which was linked to the clinical characteristics of the patient. Macrophages, neutrophils, regulatory T cells, the immunological checkpoint PD-L1, and KLHDC8A expression were all highly expressed in gliomas. According to a Cox regression study, KLHDC8A and CD68+ macrophages were predictive of poor prognosis in glioma patients. Finally, protein–protein interaction network analysis demonstrated that the expression of KLHDC8A was connected to oxidative stress and hypoxia [149].

It is possible to predict who would develop glioma using the immunological traits of macrophages, opening a new path for targeted glioma therapy. When PD1 and PD-L1 interactions were blocked in vivo, TAM phagocytosis increased, and tumour development was inhibited [150]. This finding implies that PD-1-PD-L1 therapy is appropriate for therapeutic use since it affects macrophages. In glioma cells, oxidative stress increases macrophage infiltration [151,152]. However, as was already mentioned, the most lethal primary brain tumour is GBM, which has not responded well to treatment. The specific immunological milieu of the brain is reflected by a complex tumour immune microenvironment (TIME) in GBM. Nuclear receptor subfamily 4 group A member 2 (NR4A2)-dependent transcriptional activity was induced in microglia as a result of the extreme oxidative stress. Genetic targeting of either heterozygous Nr4a2 (Nr4a2+/−) or microglia-specific Nr4a2 (Nr4a2fl/flCx3cr1cre) changed microglia plasticity in vivo by lowering alternatively activated microglia and increasing CD8+ T cells’ ability to present antigens in GBM. Squalene monooxygenase (SQLE) was triggered by NR4A2 in microglia to disrupt the homeostasis of cholesterol. Pharmacological NR4A2 inhibition decreased pro-tumorigenic TIME, and targeting NR4A2 or SQLE increased the therapeutic efficacy of immune checkpoint blockade in vivo [153]. As a result, oxidative stress and NR4A2-SQLE activation in microglia speed up the growth of tumours, which has an impact for emerging immunotherapy concepts for brain cancer.

### 2.6. Pancreatic Adenocarcinoma

Pancreatic adenocarcinoma (PAAD) is the sixth most prevalent cause of cancer-related death globally [154,155]. Pancreatic cancer patients have a poor prognosis over the long term, with a median survival time of less than six months and a five-year survival probability of only 5% [156].

Immune checkpoint therapies that target PD1/PD-L1 and CTLA-4 have recently been rapidly developed as cancer treatment options. With clinical trials demonstrating suboptimal outcomes and a poor response to PD-1/PD-L1 inhibition monotherapy, pancreatic cancer has been demonstrated to be among the most immunotolerant forms of tumours [157,158]. In a phase-2 clinical trial, 3.0 mg/kg of the anti-CTLA-4 drug ipilimumab was ineffective in treating either locally advanced or metastatic pancreatic cancer [159].

The “ConsensusClusterPlus” program was used in a study to identify molecular subtypes of pancreatic cancer based on 184 immunological markers, and the correlation between clinical characteristics and immune cell subtype distribution was examined [160]. Additionally, the correlation between immune checkpoint expression and immunological subtype composition was evaluated. For the purpose of comparing the immunological scores of various molecular subtypes, the CIBERSORT algorithm was developed. The TIDE algorithm was used by the authors to evaluate the potential therapeutic impact of immunotherapy therapies on single-molecule subtypes. Additionally, the core module of the index and its distinctive genes were identified using weighted correlation network analysis, which was used to generate the oxidative stress index using linear discriminant analysis DNA (LDA). Three molecular subtypes of pancreatic cancer—IS1, IS2, and IS3—have significantly different prognoses among various cohorts. Immune checkpoint-associated gene expression was considerably downregulated in IS3 while being elevated in IS1 and IS2, indicating that the three subgroups respond to immunotherapy interventions differently. According to the results of the CIBERSORT study, IS1 had the highest levels of immunological infiltration, whereas the TIDE analysis revealed that IS1 had a greater T-cell dysfunction score than IS2 and IS3. Moreover, compared to IS1 and IS2, IS3 was found to have a higher immunological signature index and to be more susceptible to 5-FU.

Ten possible gene markers were found using WGCNA analysis, and immunohistochemistry investigation confirmed their expression at the protein level [160]. The effectiveness of immunotherapy can be predicted by specific molecular expression patterns in pancreatic cancer, which can also affect patients’ prognoses.

In a different study, a prognostic model for PAAD was developed, and its predictive power was assessed. To find oxidative stress genes with differential expression, TCGA and three GEO datasets were employed [161]. Four genes, including MET, FYN, CTTN, and CDK1, were chosen from a list of differentially expressed oxidative stress genes in order to build a prediction model. According to GESA, the high-risk group had much more enriched immune-related pathways, metabolic pathways, and DNA repair pathways than the low-risk group. Moreover, there were noticeably more genetic alterations in high-risk groups compared to low-risk ones. In addition, the expression of synthetic driver genes for T cell proliferation and immunological checkpoint-related genes was dramatically altered, with the superior immunotherapy effect occurring in the low-risk group.

These four oxidative stress gene prognostic models may hold promise for prognostic prediction and efficacy monitoring in clinical personalized therapy. Additionally, the findings revealed that a number of immune checkpoint genes, including CD47, TNFSF9, and PVR, were significantly overexpressed in patients with high-risk scores, suggesting that this model based may serve as a guide for patients with PAAD who need individualized immunotherapy [161].

### 2.7. Hepatocellular Carcinoma

With an annually rising incidence rate, hepatocellular carcinoma (HCC) is a frequent cause of cancer-related death [162] as there is not a commonly acknowledged ideal treatment for HCC yet. Due to treatment resistance and tumour recurrence, the median five-year survival rate for HCC patients is less than 20%. In order to increase the overall survival of HCC patients, it is imperative to discover new treatment agents.

The existence of a strong association between the redox system and HCC has been shown by numerous experiments. In an in vitro study, the flavonoid calycosin-7-glucoside (CG) inhibits the growth of human cancer cells by targeting Trx1 [163]. The effects of Trx1 on the CG-induced inhibition of HCC were then further investigated using si-TRX1. According to the findings, CG dramatically increased OS, significantly increased apoptosis, decreased Huh-7 and HepG2 cell growth, and suppressed Trx1 expression. Moreover, in vivo studies shown that CG dose-dependently controlled the expression of Trx1, oxidative stress, and apoptotic proteins to slow the growth of HCC [163].

Thus, a connection between immunological therapy and OS was displayed, and other studies confirmed this relationship. Nuclear factor erythroid 2 like 1 (NFE2L1/Nrf1) is a crucial component of the cap’n’collar basic-region leucine zipper (CNC-bZIP) family of antioxidant transcription factors [164]; in liver cancer tissues, NFE2L1 expression is very low or non-existent, and it is correlated with the clinical stage of liver cancer [165,166,167,168]. Animal studies demonstrated that, once the NFE2L1 gene was deleted from the liver, all mice developed HCC without further stimulation [166]. According to research, TNFSF15 is a specific downstream gene of NFE2L1, which has the ability to regulate TNF expression. Furthermore, liver cancer tissues express both NFE2L1 and 41BBL at low levels [168], suggesting that 41BBL may be related to the formation of HCC brought on by NFE2L1 deletion. Therefore, it is possible that NFE2L1 can regulate the transcription of 41BBL. According to transcriptome data, 41BBL might be an immunological checkpoint that responds to OS. The outcomes of the experiment on promoter activity demonstrated that NFE2L1 can activate the transcription of the 41BBL gene via the ARE component in the promoter region. Furthermore, overexpression of 41BBL has been linked to both cell senescence and cell growth, according to cell biology research. Significantly, ROS levels in the cells greatly rose after 41BBL was overexpressed, whereas NFE2L1 was repressed. This suggests that 41BBL has the ability to regulate OS in the cells through feedback. Thus, a key mechanism that mediates the interaction between oxidative stress and the tumour immune response may be represented by the NFE2L1/41BBL axis [169].

However, the relationship between OS and the effectiveness of check point inhibitors in HCC has been supported by several investigations. While adoptive transfer of Tet2 -deficient B cells inhibited the growth of HCC, OS from the HCC microenvironment activated ten-eleven translocation-2 (TET2) in B cells, promoting IL-10 production. TET2 must bind to the aryl hydrocarbon receptor in order to hydroxylate IL-10. Moreover, patients with HCC have a bad prognosis if their B cells have high amounts of IL-10, TET2, and 5hmc. Additionally, a study used 5hmc to measure TET2 activity in B cells to assess the effectiveness of anti-PD-1 therapy. Particularly noteworthy is that TET2 inhibition in B cells promotes antitumour immunity to enhance anti-PD-1 therapy for HCC [170].

### 2.8. Gastric Cancer

One of the main causes of cancer-related death has been gastric cancer (GC) [171]. Because stomach cancer is typically already advanced when it is discovered, the mortality rate was 8.2% in 2018 [105]. When diagnosed, the majority of GC patients have advanced disease. There is a critical need for innovative therapeutic approaches because chemotherapy or surgery alone frequently produces subpar outcomes.

In assessing the effectiveness of immunotherapy in this kind of patient, the redox state analysis appears to be crucial. After thorough molecular analysis of 295 primary gastric cancers, the TCGA database identified four different subtypes: Epstein–Barr virus (EBV), microsatellite instability (MSI), chromosomal instability (CIN), and genome stable (GS) [172,173,174]. As immunotherapy advances, recognition of molecular subtypes can assist in establishing a new paradigm of cancer therapies. Yet, the therapeutic efficacy of immunotherapy varied according on the molecular subtype. Interestingly, solid tumours with the MSI phenotype responded to anti-PD1 drugs more dramatically than solid tumours with the Microsatellite Stable (MSS) phenotype [175,176,177]. In contrast to GS and CIN, metastatic GC patients with the MSI and EBV subtype responded remarkably to PD1 inhibitors and treatment [178,179]. The correct evolution of EBV infection and MSI status could therefore be exploited as a possible biomarker for anti-PD1/PDL1 targeted therapy in GC. The High-Microsatellite Instability (MSI-H) phenotype has gained significant recognition as a prognostic indicator for immunotherapy as a result of the high PD-L1 expression [180,181,182].

The correlation between EBV, d-MMR/MSI-H subtypes, and clinical characteristics in GC cases was thoroughly summarized by Cai et al. [183]. Additionally, using bioinformatics techniques, the GSE62254/ACRG and TCGA-STAD datasets, which came from GEO and TCGA, respectively, were examined to identify the genetic features associated with prognosis. Regardless of race, the clinical examination of the GSE62254/ACRG and TCGA-STAD datasets revalidated the positive predictive significance of MSI in various cohorts. Subsequently, using weighted gene co-expression network analysis, a critical gene module that was strongly related with improved status and a longer overall survival duration for MSI cases was discovered [183].

Additionally, it is well-recognized that oxidative stress can initiate autophagy. Yao et al. attempted to investigate novel autophagy-related clusters and create a multi-gene signature for GC that could predict prognosis and immunotherapy response [184]. Using consensus clustering, a total of 1505 individuals from eight GC cohorts were divided into two subgroups. Patients in cluster 1 have better survival rates and epithelial-mesenchymal transition scores than those in cluster 2, which is the comparison group. High heterogeneity in terms of immune cell infiltration, somatic mutation pattern, and pathway activity via gene set enrichment analysis are further characteristics of the two subtypes. PTK6 amplification and BCL2/CDKN2A deletion were found in a majority of the 21 autophagy-related differential expression genes (DEGs) identified by the authors. The four-gene risk signature (BNIP3, HSPB8, GABARAPL1, and PEA15,) was further built and validated in three separate datasets with strong predictive performance. The risk score has been shown to be a reliable prognostic indicator. Strong validity of GC survival was revealed by a prognostic nomogram. Immune cell infiltration status, tumour mutation burden, MSI, and immune checkpoint molecules were all substantially correlated with the risk score. The IMvigor210 cohort proved the model’s accuracy in predicting the outcome of immunotherapy and tumour-targeted therapy [185]. Additionally, this model can distinguish between patients with low and high risk.

### 2.9. Colorectal Cancer

Crosstalk between oxidative stress and ferroptosis is shown in a variety of human illnesses, including colorectal cancer (CRC). To predict the prognosis and therapy response in CRC patients, an experiment was conducted to generate an oxidative stress- and ferroptosis-related gene (OFRG) prognostic signature [185]. As OFRGs, 34 insertion genes between genes associated with OS and genes related to ferroptosis were found. Three OFRG clusters were created from patients with CRC, and DEGs (differentially expressed genes) between clusters were found. OFRG clusters were linked with both immune cell infiltration and patient survival. Patients in the low-risk group in this study had better prognoses, higher levels of immune cell infiltration, and better reactions to fluorouracil-based chemotherapy and immune checkpoint blockade therapy than patients in the high-risk group. Hence, hot tumours might be used to describe CRC at low risk, whereas cold tumours could be used to describe CRC at high risk. The expression levels of five hallmark genes in CRC and nearby normal tissues were further confirmed using an in vitro experiment in order to further uncover possible biomarkers for CRC [185]. In conclusion, OFRG-related prognostic signature performed exceptionally well in predicting survival and treatment outcomes for CRC patients treated with immune checkpoint inhibition. For specific treatment methods in practical practice, this might be useful.

### 2.10. Renal Cancer

Although other kidney cell types have also been proposed as the primary source of renal cell carcinoma (RCC), tubular cells are the primary source of this malignancy [186]. More than 400,000 new cases, or 2% of all cancer diagnoses, were reported in 2020 according to GLOBOCAN data, with a higher prevalence in male patients [187].

RCC treatment options have changed significantly over time. Although their inherent toxicity, IL-2 and IFN-based therapies were the most widely used choices over two decades ago. Sunitinib, Bevacizumab, and subsequent combinations with PD-1 inhibitors such as Pembrolizumab were used to target the VEGF angiogenic pathway [188]. In fact, both ICI and ICI plus TKI are increasingly being investigated as treatments for RCC.

Because RCC are among the most immune-infiltrated tumours, therapeutic methods based on ICI are quite pertinent [189,190]. It is interesting to note that mounting data point to a strong correlation between the activation of metabolic pathways and angiogenesis and inflammatory markers [191,192]. Therefore, in silico analysis revealed that both the metabolic and immunological status of the tumours might be utilized to predict prognosis [193]. High metabolic activity in RCC tumours has been shown to decrease immune infiltration. According to research [194], patients with RCC who exhibit high inflammation and low metabolic activity benefit most from immunotherapy, which is consistent with this idea and the finding that the tumour microenvironment has a significant impact on responses to systemic therapy [195].

The activation of the NRF2 pathway and other ROS-producing drugs has been associated with resistance to inducers of ferroptosis, a regulated form of iron-dependent oxidative cell death [196]. Translocation renal cell carcinoma (tRCC), a poorly understood subtype of kidney cancer, is mostly brought on by MiT/TFE gene fusions. A study [197] identified the hallmarks of tRCC by an integrated analysis of 152 patients with the disease who were found in genomic, a clinical trial, and retrospective cohorts. With the exception of MiT/TFE fusions and homozygous deletions at chromosome 9p21.3, the majority of tRCCs have minor somatic changes. A stronger NRF2-driven antioxidant response is observed transcriptionally in tRCCs, and this response is linked to resistance to targeted treatments.

Results for subjects with tRCC cured with vascular endothelial growth factor receptor inhibitors (VEGFR-TKI) were consistently reported to be poorer than those for those treated with ICI. While the tumours are infiltrated with CD8+ T cells, the data from multiparametric immunofluorescence showed that these T cells have an exhaustion immunophenotype that is different from that of clear cell RCC. These findings contributed to a thorough understanding of the clinical and molecular aspects of tRCC and may inspire new therapy ideas.

Similar results were attained by Ren et al. who conducted a predictive signature study in patients with Kidney Renal Clear Cell Carcinoma (KIRC) [198]. Authors examined the gene expression and clinical information of KIRC patients with the aid of the TCGA database. The identification of antioxidant-related genes with notable variations in expression between KIRC and normal samples followed. Patients with higher risk scores had a worse prognosis, more advanced grade, and stage, and more M0 macrophages, regulatory T cells, and follicular helper T cells than other patients. Between the two risk groupings, there were statistically significant variations in the expression of the HLA and checkpoint genes. In various risk groupings, the study examined the expression levels of 12 immune checkpoint genes. The research on the association between immune checkpoint gene expression and risk score offered a fresh perspective on immunotherapy. The expressions of BTLA, CD137, CD27, CD276, CD28, CTLA4, HCVCR2, LAG3, PD1, TNFRSF4, TNFRSF18, and TNFSF14 were higher in the high-risk group compared to the low-risk group, suggesting that the prospective immune checkpoint inhibitors may have an impact on high-risk KIRC patients. The nomogram performed well and accurately predicted the 3-year and 5-year survival of KIRC patients. The different set-up could justify the different therapeutic success obtained in these patients by the immunotherapy treatment [198].

### 2.11. Oral Squamous Cell Carcinoma

Almost 90% of oral cancers are caused by oral squamous cell carcinoma (OSCC), which is sixth in the world in terms of cancer incidence [199]. Although medicines have made great improvement, the overall survival rate is still pitiful at 20% [200].

Based on mRNA expression data from the TCGA database, Lu et al. created an oxidative stress-related prognostic signature and assessed its relationships with OSCC prognosis, clinical characteristics, immunological status, immunotherapy, and medication sensitivity through a number of bioinformatics analysis [201]. The signature was demonstrated to be an independent prognostic factor with high accuracy and was proven to be a good indicator for predicting the prognosis and immunological state of patients with OSCC on the basis of both the TCGA-OSCC and GSE41613 cohorts. Additionally, they discovered that the risk score had a substantial impact on the chemotherapeutic sensitivity and tumour microenvironment and that immune checkpoint therapy would be more advantageous for patients with high-risk scores than for patients with low-risk scores. The prognostic signature might offer a reliable and effective predictive tool that might forecast prognosis and immunological state and direct clinicians in creating individualized treatment plans for patients with OSCC [201].

### 2.12. Lung Adenocarcinoma

One of the most frequent malignant tumours and the main reason for cancer-related death worldwide is lung cancer, and surgery can only be used to treat up to one-third of patients [202].

To study the effect of OS on the prognosis of patients with lung adenocarcinoma (LUAD), Zhu et al. created a prognostic risk score model and tested its predictive power in the TCGA and GEO cohorts [203]. To further examine the probable processes in LUAD, they divided the patients into two groups and then carried out analyses of immune cell infiltration, mutational landscape, immunological checkpoints, and correlation of treatment response. They discovered a prognosis-related risk model based on a LUAD gene signature for OS, which includes the genes CYP2D6, FM O 3, CAT, and GAPDH.

As comparison to the low-risk group, LUAD patients in the high-risk group had a shorter overall survival. Lung cancer cells’ ability to proliferate, invade, and migrate may be reduced by overexpressing CAT. Tumour-associated immune cell infiltration and immune checkpoint molecule expression were substantially correlated with each other, and the tumour mutation burden was higher in LUAD patients with high-risk scores. Between the high- and low-risk groups, there are considerable differences in drug sensitivity, which may have an impact on clinical treatment decisions and provide new knowledge for future anti-tumour immunotherapy [203].

A different study performed an integrative analysis of genomic, transcriptomic, and proteomic data from early stage and chemo-refractory KRAS-mutant lung adenocarcinoma (LUAC) and discovered three strong KRAS-mutant LUAC subgroups, each of which was dominated by co-occurring genetic events in STK11/LKB1 (the KL subgroup), TP53 (the KP subgroup), and CDKN2A/B inactivation along with low expression of the NKX2-1 (TTF1) transcription factor (the KC subgroup) [204].

KL tumours showed fewer immunological markers, such as PD-L1, and had high frequencies of KEAP1 mutational inactivation. Inflammatory indicators, immunological checkpoint effector molecules, somatic mutation levels, and relapse-free survival were all higher in KP tumours. Drug sensitivity patterns varied, and KL cells particularly exhibited a greater sensitivity to HSP90-inhibitor therapy. This study offers proof that KRAS-mutant LUAC subgroups with unique biology and treatment vulnerabilities can be identified by co-occurring genetic changes [204].

Lastly, earlier research created helical polypeptides that can target the mitochondria and trigger mitochondria-dependent apoptosis [205]. The mitochondrial membranes may be targeted and destabilized by the helical polypeptides, which also enhanced OS and programmed cell death. Because of the development of intracellular oxidative conditions, their ability to damage mitochondria is likely to be used as an ICD inducer. Stimulating the cytotoxic T cell response and decreasing the immunosuppressive tumour microenvironment, fluorinated mitochondria-disrupting helical polypeptide (MDHP) and antiPD-L1 dramatically inhibited tumour development in in vivo experiments and prevented metastasis to the lungs. Fluorinated MDHP and PD-L1 combined cancer immunotherapy significantly reduced tumour growth and lung metastasis. This treatment reduced the number of immune suppressor cells such as MDSCs and Tregs while triggering immunological responses against a tumour. These findings imply that immune checkpoint blockade treatment and fluorinated MDHPs, an ICD inducer, work synergistically to provide an effective cancer immunotherapy regimen [206] (Table 1).

## 3. Conclusions and Future Perspectives

The tumour microenvironment, which has intricated biological and chemical linkages and offers a favourable substratum for tumour progress and expansion, significantly contributes to the advancement of cancer. Targeting tumour cells while disregarding the contiguous TME is ineffective in curing cancer, according to mounting research. TME contributes to the immune escape of tumours via a variety of direct and indirect mechanisms on particular immune cells, including the generation of various cytokines/chemokines. interactions that influence the activation of regulatory immune cells, as well as the reprograming of an immunosuppressive function in immature myeloid cells. One of the primary regulatory mechanisms that inactivates tumour-infiltrating lymphocytes in cancer lesions is the activation and overexpression of inhibitory immunological checkpoints or their ligands in TME compartments. In fact, according to their targets and modes of action, ROS production can cause both an up- and a down-regulation of PD-L1 expression in cancer cells [207].

In fact, different studies have pointed out a regulation of PD-L1 expression by ROS. Moreover, in the tumour microenvironment, high levels of intracellular ROS can stimulate the secretion of IL-6, a pro-inflammatory cytokine that facilitates proliferation, migration, and metastasis of several types of tumours [208,209], and the anti-tumour immune response to anti-PD-L1 treatment in solid tumours has been demonstrated to be significantly suppressed by IL-6 [210,211]. Therefore, one of the essential approaches to comprehend the function of the immune response in cancer patients undergoing ICB therapy is to investigate the particular involvement of ROS in the PD-1/PD-L1 pathway across cancer types. Thus, altering the redox system could be a valuable strategy to change the anti-cancer effectiveness of immunotherapy, and new tactics have been developed in addition to the aforementioned approaches in order to get better outcomes, such as the employ of vitamin C.

Humans and the majority of animals are unable to synthesis vitamin C, making it a vital natural micronutrient and a significant physiological antioxidant [212]. Vitamin C is well-known to help prevent a variety of disorders, particularly those brought on by oxidative stress, including cardiovascular and neurological dysfunctions [213,214,215]. As for neoplastic diseases, it is interesting to note that vitamin C has been demonstrated to have the ability to eradicate different cancer cell types in vitro [216,217,218]. Moreover, a number of studies have shown that supplementing cancer cell lines with high doses of vitamin C with standard anti-cancer medications increased their cytotoxicity [219,220,221,222,223].

The anti-tumour properties of vitamin C have been attributed to several molecular mechanisms, interactions that modify the immunosuppressive role of immature myeloid cells and the activation of regulatory immune cells [224]. The activation and overexpression of inhibitory immunological checkpoints or their ligands in TME compartments is one of the main regulatory mechanisms that inactivates tumour-infiltrating lymphocytes in cancer lesions. In fact, PD-L1 expression in cancer cells can be up- or down-regulated by ROS generation, depending on their targets and modes of action [207].

It is interesting to note that a recent study found that vitamin C regulates epigenetic processes via DNA demethylation, which may be related to its anti-cancer effects [225]. Moreover, it is significant to note that vitamin C can influence the TME by promoting T lymphocyte invasion and cytokine production. As a result, vitamin C may be crucial in the control of the immune system’s anti-tumour response. In a syngeneic lymphoma mouse model, it has been demonstrated that vitamin C in large doses enhanced the anti-tumour impact of anti-PD-1 therapy [226]. Furthermore, a recent study found that high doses of vitamin C can enhance the cytotoxic activity of adoptively transferred CD8+ T cells, modulate immune cells’ infiltration into the tumour microenvironment, and work in conjunction with anti-PD-1 and anti-CTLA-4 checkpoint inhibitors in mice bearing syngeneic tumours [227]. Vitamin C has been shown to boost the intra-tumoral infiltration of CD4+ and CD8+ T lymphocytes, as well as macrophages, into the cancer milieu by increasing the production of granzyme B and interleukin-12. High doses of vitamin C have also been demonstrated to cooperate with anti-PD-1 and anti-CTLA-4 therapies in mice bearing syngeneic malignancies, alter immune cell infiltration into the tumour microenvironment, and increase the cytotoxic activity of adoptively transplanted CD8+ T cells [227]. It is intriguing to note that vitamin C may support anti-PD-L1 activity by increasing levels of the chemokines CXCL9, CXCL10, and CXCL11 and promoting lymphocyte invasion of tumours while boosting anti-tumour immunity [228].

However, it is crucial to highlight a few factors that make vitamin C’s effect specific to certain neoplastic disease types. The expression of sodium-dependent vitamin C transporters 1 and 2 (SVCT1 and SVCT2) is linked to the cytotoxicity of vitamin C in the tumour microenvironment. In fact, vitamin C treatment of cancer cells with high SVCT-2 expression levels led to the observation of a sizable cytotoxic effect. Only high dose vitamin C (>1 mM) had an anti-cancer impact in cell lines with low SVCT-2 expression, but low-dose therapy led to the proliferation of cancer cells. Additionally, SVCT-2 may be used as a marker for the start of vitamin C therapy, according to a study on breast cancer. The same study also shown that SVCT2 knockdown in breast cancer cells led to resistance to vitamin C therapy [229]. These findings help to understand why vitamin C therapy does not have the same positive effects on cancer patients. However, although there is mounting evidence that vitamin C may improve immunotherapy response, more pre-clinical and clinical research is still required to fully understand and confirm this impact [227].

Many other pharmaceutical strategies have been tested. Currently being examined in clinical trials as an anticancer medication is auranofin (AUR), a medication already approved for the treatment of rheumatoid arthritis. The medication causes a significant amount of oxidative stress, which leads to the ROS-mediated inhibition of enzymes such hexokinase [230]. AUR has recently been discovered to significantly increase the expression of PD-L1 on cancer cells and to promote CD8+ T-cell infiltration in the tumour [83]. AUR boosted the expression of PD-L1 at the surface of cancer cells both in vitro and in vivo, favouring tumour resistance in breast cancer cells. When coupled with an antibody that targets PD-L1, it increased PD-L1 presumably via the Erk1/2-Myc pathway and demonstrated synergistic effect [83]. In this interesting study, it was shown that auranofin was extremely efficient in causing cell death and preventing the growth of triple negative breast cancer cells grown as spheroids. It was also shown that AUR was effective in vivo in patient-derived tumour xenografts by inhibiting TrxR1 activity and increasing CD8+T cell infiltration.

AUR and an anti-PD-L1 antibody together synergistically reduced the growth of syngeneic 4T1.2 primary tumours [231]. Moreover, in vivo hepatocellular carcinoma treatment with AUR and anti-PD-L1 therapy was successful in enhancing the ICD effects of oxaliplatin [232,233]. This possibility is further bolstered by research that demonstrates the sensitivity of breast cancer tissues to the antiprogestin and antiglucocorticoid drug mifepristone [234], which can induce ICD and work in synergy with PD-L1 inhibition [235]. According to a recent study, mifepristone’s anti-cancer actions were linked to the generation of proteotoxic ER stress, a route that must be activated for ICD to be effective [236].

Butaselen, a TrxR1 inhibitor and cellular ROS producer that may be helpful in the chemoprevention of hepatocellular carcinoma, has been shown to have a very similar effect. Via the STAT3 pathway, butaselen was discovered to inhibit PD-L1 expression on the surface of tumour cells [237].

The anticancer compound chaetocin, a natural product obtained from the fungal species *Chaetomium*, was discovered to follow the similar pattern. Inducing an excessive build-up of ROS in cells, chaetocin is a powerful TrxR1 (and histone methyltransferase) inhibitor that causes cancer cells to undergo apoptosis [238]. In human pancreatic cancer cells treated with chaetocin, it was discovered that the amount of the PD-L1 protein had significantly decreased [239]. It is feasible that a change in PDL-1 levels could affect how well immunological check point inhibitors work as a treatment.

Trifluoperazine (TFP), an antipsychotic, and disulfiram (DSF), a medication used to treat chronic alcoholism, are two further medicines that modify ROS that are worth mentioning. The treatment of cancer with both medications is also being considered [42]. In recent studies, TFP, a phenothiazine-type calmodulin inhibitor, has been shown to raise ROS levels in colorectal cancer cells while also encouraging the production of PD-L1 in these cancer cells and of PD-1 in the tumour-infiltrating CD4+ and CD8+ T cells [240].

Recent research has shown that photobiomodulation (PBM) using near-infrared (NIR) light in the NIR-II window reduces OS and encourages CD8+ T cell proliferation, suggesting that PBM using NIR-II light may boost anti-cancer immunity [241]. A novel method to help CD8+ T cells that have penetrated tumours is to combine a high tissue penetration depth NIR-II laser with PBM, claims a study. Brief treatments with simultaneous 1064 and 1270 nm lasers reduced the expression of PD-1 in CD8+ T cells in a syngeneic animal model of breast cancer [153]. The adoptive transfer of laser-treated CD8+ T lymphocytes ex vivo against a model tumour antigen resulted in a better tumour development delay, which demonstrated the NIR-II laser treatment’s direct impact on T cells. Additional research revealed that a specific set of NIR-II laser characteristics improved the effect of an immune checkpoint inhibitor on tumour growth. PBM with NIR-II light increases the efficacy of cancer immunotherapy by encouraging CD8+ T cells. Contrary to current immunotherapy, which entails risks of negative drug–drug interactions and major adverse events, the laser is inexpensive, safe, and it may be widely combined with other therapies without modification to achieve therapeutic value.

As a therapeutic intervention, NIR-II light-based immunotherapy offers many benefits, including deep tissue penetration and no mutagenic or carcinogenic potential. The NIR laser poses little to no danger of negative effects when used properly [153].

Carbon monoxide (CO) gas therapy is also becoming more and more popular because of its incredible anticancer potential and lower side effects [242,243]. CO was discovered to be an essential biogas transmitter linked to mitochondria as an endogenous gaseous molecule. This biogas transmitter can greatly speed up mitochondrial respiration, which forces cancer cells to use more oxygen to produce energy [244,245]. These metabolic processes cause the mitochondria to be depleted and a significant amount of ROS to be produced [246,247]. Most importantly, exogenous CO possesses potent immune regulation capabilities that, in addition to its direct anti-tumour action, can affect the recruitment of innate immune cells and the growth of myeloid cells [248]. Additionally, CO has no detrimental effects on healthy cells and can destroy cancer cells by decreasing medication resistance [249,250].

For tumour control, metastasis prevention, and recurrence prevention, CO gas therapy is used in a study [251]. To overcome the drawbacks of monotherapy, it is suggested that CO2-g-C3N4-Au@ZIF-8@F127 (CCAZF) combine immunotherapy and gas therapy into a photocatalytic nanogenerator. The highly efficient CO release behaviour displayed by CCAZF causes ICD by gradually escalating the OS in tumour cells. When ICD is induced, CO therapy improves immune responses and makes it possible for effective immune cells to be activated. CCAZF exhibits an increased immunological impact when coupled with ICB, which causes the regression of primary and distant tumours [251]. This approach to in situ photocatalytic CO therapy provides a fresh approach for developing novel ICD inducers while also largely avoiding the toxicity of CO leakage (Table 2).

A biomimetic Ru-TePt@siRNA-MVs multifunctional nano-integrator was created by Wu et al. [252] and shown good specificity for the transport of siRNA, exogenous US irradiation, and endogenous TME. In a nutshell, this technology has the ability to effectively control TME hypoxia and then combine gene immunotherapy with ROS to increase anticancer efficacy. In addition to their sonodynamic and chemical kinetic effects, the Ru-TePt spindle nanorods’ plentiful positive charges on the surface enable effective siRNA loading. Moreover, adaptable nanotheranostics were created by modifying cell membrane vesicles that expressed transferrin (MVs-Tf). By RNA interference gene suppression of the PD-1/PD-L1 pathway, tumour-targeting Ru-TePt@siRNA-MVs can enhance ROS-induced cancer immunotherapy.

This layout has several clear advantages. The charge-trapping properties of Ru-TePt nanorods prevent the recombination of electron-hole(e−–h+) pairs in a way that triggers an adaptive immune response and causes immunogenic cell death. Additionally, the encapsulation of biocompatible cell membrane vesicles can further improve NRs targeting and endosomal escape, resulting in promising gene delivery efficacy and sonodynamic therapy effectiveness. The chemical kinetic effect of Ru-TePt nanorods can also produce toxic OH, amplifying ROS-based cell killing. Moreover, in the presence of immunological checkpoints blocked cytotoxic T cells, enhanced cytotoxic ROS efficiently induces ICD to enhance anti-tumour immunity.

Other methods will soon be able to improve the PD-L1 control mechanisms, increasing the potential for treating neoplastic diseases. Clustered regularly interspaced short palindromic repeats/CRISPR-associated nuclease 9 (CRISPR/Cas9) gene editing technology, which can recognize the target genome sequence with single-strand guide RNA (sgRNA) and direct the Cas9 protease to knock down the target gene, has significantly aided the development of anti-PD-1/PD-L1 tumour immunotherapy [253]. The TME’s lymphocytes can be functional again thanks to the CRISPR/Cas9 technology, which can boost PD-L1 deletion at the genomic level and diminish PD-L1 expression. Zhang et al. combined focal adhesion kinase (FAK) siRNA, Cas9 mRNA, and PD-L1 sgRNA into self-assembling lipid nanoparticles (LNPs) to overcome the rigid extracellular matrix and PD-L1 overexpression. Gene editing was increased >10-fold in tumour spheroids as a result of enhanced cellular uptake and tumour penetration of nanoparticles brought on by FAK-knockdown. In four mouse cancer models, including those with ovarian and liver cancer, the addition of siFAK + CRISPR-PD-L1-LNPs significantly reduced tumour growth and metastasis [254]. A study found that the deletion of TP53 resulted in reduced ex vivo macrophage phagocytic activity, which is a common cause of chemoimmunotherapy resistance in B-cell malignancies. Additionally, TP53 deletion boosted the expression of PD-L1 and the production of extracellular vesicles by B-cell lymphoma cells. The improvement of macrophage phagocytic capacity and in vivo therapeutic responsiveness by PD-L1 CRISPR-KO was due to the disruption of EV-bound PD-L1 [255].

Finally, a variety of other mechanisms that can control PD-1 expression should be investigated. The TRAF family of proteins includes the protein known as TRAF6. In response to signals from the TNFR and interleukin-1 receptor/TLR superfamilies, it encourages the activation of the transcription factors NF-kB and AP-1, both of which cause the production of pro-inflammatory cytokines in myeloid cells. TRAF6 signalling in T cells also controls how they develop and function. It was discovered that immunological checkpoint molecules CTLA-4 and PD-1 are expressed in considerably larger concentrations on the cell surfaces of T cells with defects in the TRAF6 gene. These findings show that the TRAF6 signalling pathway in T cells, which controls anti-tumour immunity, activates the tumour-specific CTLs and Th9 cells in a tumour microenvironment [256]. A mechanistic study evaluated the efficacy of combining ATR inhibitors (ATRi), irradiation (IR), and anti-PD-L1 antibodies was evaluated in colorectal cancer. The authors showed that IR + ATRi could attenuate SHP1-mediated inhibition of the TRAF6-STING-p65 axis by promoting SUMOylation of SHP1 at lysine 127, thereby activating both the canonical cGAS-STING-pTBK1/pIRF3 axis and the non-canonical STING signalling. Immunotherapy was made easier by IR + ATRi, which increased STING signalling, produced type I interferon-related gene expression, robust innate immune activation, and revived the cold tumour microenvironment. Thus, the combination of ATRi and IR could facilitate anti-PD-L1 therapy by promoting STING signalling in CRC models with different microsatellite statuses [257].

Finally, a number of studies have suggested that angiogenesis is crucial to the pathophysiology of cancer [258,259,260]. Analysing the impact of immunotherapy and antiangiogenic drug combinations in cancer patients would be helpful given that both treatments may have synergistic antitumor effects. Anlotinib and PD-1 inhibitors together are a potential therapy option for LUAD patients who have developed EGFR-TKI resistance, according to a study [261]. Recent research has also demonstrated the strong relationship between angiogenic dynamics and oxidative stress [262], and it is possible to interfere with this axis using drugs that might alter the generation of ROS, hence enhancing the efficacy of immunotherapy.

In conclusion, the in vitro and in vivo experimental findings showed that several strategies might be used to successfully activate the immunological response triggered by OS and suppress immune resistance mediated by the PD-1/PD-L1 axis. As a result, this coordinated therapeutic paradigm offers insightful information for creating prospective oxidative stress and genetic immunotherapy. The use of metabolic modulators, particularly those that may target oxidative metabolism, to promote immunotherapeutic response is supported by literature evidence. Normalizing the oxygen tension of the tumour microenvironment is an interesting method for increasing immunotherapy’s effectiveness. The use of oxidative stress modulators may be able to alter the tumour microenvironment so that it is metabolically conducive to T cell function, thereby increasing the effectiveness of immunotherapeutic cancer treatments.

## Figures and Tables

**Figure 1 biomedicines-11-01325-f001:**
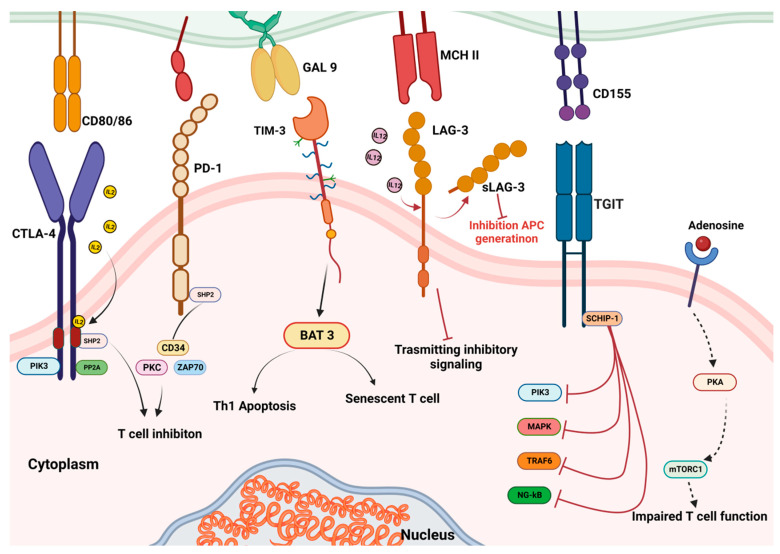
Immune checkpoint structure and interactions.

**Figure 2 biomedicines-11-01325-f002:**
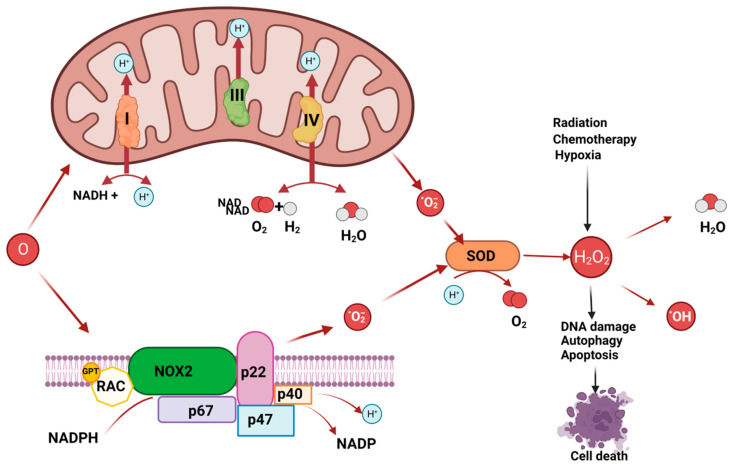
ROS production may derive by mitochondrial respiratory chain and NADPH oxidase or may derive from hypoxia, or chemotherapy. ROS may induce DNA base modifications, DNA damage and cell death.

**Figure 3 biomedicines-11-01325-f003:**
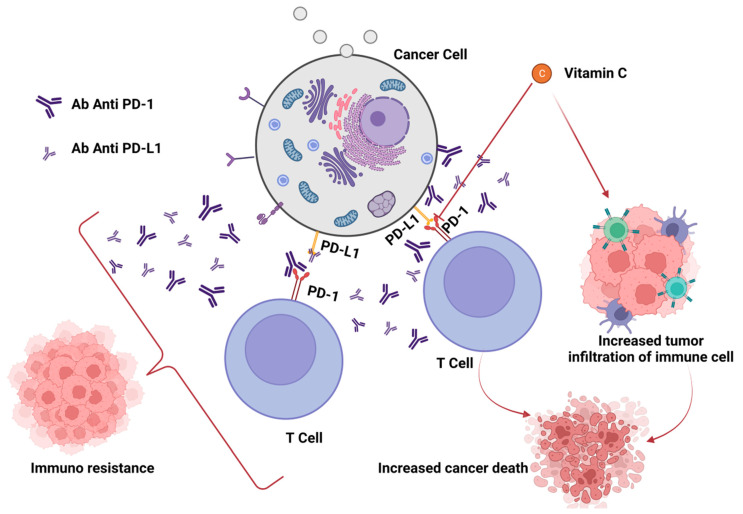
Effects of vitamin C on efficacy of checkpoint blockade treatment.

**Table 1 biomedicines-11-01325-t001:** Modification of the oxidative state and its consequences on the immunotherapy treatment.

Disease	Oxidative Status	Mechanism	Intervention	Ref.
Breast cancer	Thioredoxin pathway genes overexpression	Reduced ROS production; increase CD8 tumour infiltration; up regulation of PD-L1 expression	Auranofin Anti PD-L1 antibody	[83]
Ovarian cancer	Increased ROS levels	High oxidative stress-related risk score patients have minor benefits from immune treatment	Anti PD1/L1 immunotherapy	[104]
Endometrial cancer	Solute carrier family 7-member 11 overexpression	Immune cell infiltration; correlation with PD-1, PD-L1, PD-L2 and CTLA-4 expression	Salubrinal, S-trityl-L-cysteine	[113]
Melanoma	Reduced glutathione content; modified KEAP1-NRF2 and glyoxalase 1 pathways	TXNIP-dependent phenomena	GLO1 inhibitor	[128]
Glioma	Increased ROS production	KLHDC899 overexpression; increased macrophage infiltration	Anti-mouse PD-1 specific monoclonal antibody	[153]
Pancreatic adenocarcinoma	Increased oxidative stress genes expression	Changed expression of immune checkpoint genes	Immunotherapy	[161]
Hepatocellular carcinoma	Modified TRX1 levels	Effects of 41BBL and NFE221 levels; TET2-medated IL-10+ B-cell generation	Bobcat229, anti PD-1	[170]
Gastric cancer	Oxidative phosphorylation and glutathione changes	PTK6 amplification, BCL2/CDKNA deletion, modified autophagy	Immunotherapy	[184]
Colorectal cancer	Modified oxidative stress	Ferroptosis, reduced immune infiltration	PD-1/PD-L1/PD-L2 blockade	[185]
Renal cancer	NFR2 antioxidant response change	Expression of T effectors	Immunotherapy	[197,198]
Lung carcinoma	Increased oxidative stress		Anti PD-L1 and Fluorinated mitochondria-disrupting helical polypeptide	[206]

**Table 2 biomedicines-11-01325-t002:** Possible interventions to increase the effectiveness of therapy with check point inhibitors.

Molecule	Disease	Mechanism	Treatment	Study	Ref.
Vitamin C	Lymphoma	Increased CD8+T cells	Anti PD-1 therapy, anti CTLA-4	In vivo	[226,227]
Auranofin	Breast cancer	Modified ROS amount, inhibition of hexokinase, increase of CD8+T cell infiltration, inhibition of TrxR1 activity, proteotoxic ER stress	Anti-PD-L1, anti PD-L1 plus cisplatin	In vivo and in vitro	[83,231,234]
Butaselen	Hepatocellular carcinoma	TrxR1 inhibition, inhibition of PD-L1 expression		In vitro	[237]
Chetocin	Pancreatic cancer	TrxR1 inhibition, reduction of PD-L1		In vitro	[239]
Photo-biomodulation and Near-infrared light	Breast cancer	Reduction of PD-1 in CD8+ T cells, CD8+ T cell proliferation.	Immune check point inhibitors	In vivo	[153]
Carbon monoxide	Cancer cells	ROS modification	Immunotherapy	In vitro	[251]

## Data Availability

Not applicable.

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
