# Peer review of "Redox Signaling Modulates Activity of Immune Checkpoint Inhibitors in Cancer Patients"

_biomedicines, 2023, doi:10.3390/biomedicines11051325_

Round 1

Reviewer 1 Report

Review manuscript entitled " Redox signaling modulates activity of immune checkpoint inhibitors in cancer patients" is primarily a important piece of information for cancer diagnostic and information provided has multiple application in cancer clinical application. Manuscript can be accepted with following condition

1) Avoid repeated small paragraphs of 3-5 line. and also do not mix the concept of different themes in one paragraph.

2) More explanation is required how SCHIP-1 blocks Kinases pathway and TRAF6 and NG-kB pathway.

3) How Vit.C acts on PD-1 PDL1 to activates T cell which intern increase the cancer death, IS it cancer death or cancer cell death. And is it universal phenomenon or particularly to certain type of cancer.

4) Conclusion is not sufficient. More focused and crisp information is required.

Author Response

Dear reviewer, I revised the paper accordingly to your suggestions

Giuseppe Murdaca

Reviewer 2 Report

The manuscript is a comprehensive review presenting redox modulation of  activity of immune checkpoint inhibitors in various types of cancer.

The effect of ROS-inducing compounds on the PD-L1 expression is complex. Can the effects be due to other ways of action, apart from ROS induction?

Line 19: “Treg”, though the acronym is in common use, please explain it for full clarity

Line 32: should be ”programmed death receptor”

Line 52: “mAbs”, though the acronym is obvious, please explain it on the first use

Line 72: is “the oncogene” optimal?

Figure 2 might be more precise. NADPH oxidase, mentioned in the legend, is not presented. H+ are released primarily into the intramembrane space, not outside the outer membrane. Do the antioxidant enzymes produce H2O2? It is true only for superoxide dismutase.

Line 117: “organoselen”,please change to “organoselenium”

Lines 117/118: “is inhibited by the organoselen chemical ethaselen (BBSKE), which is frequently overexpressed in a variety of cancer types”, the enzyme is overexpressed, not BBSKE, please make it more clear

Lines 20/121; “BBSKE's suppression of TrxR1 causes cells to produce more ROSBBSKE's suppression of TrxR1 causes 120 cells to produce more ROS”; is it increased production or inhibition of scavenging?

Line 145:” cathecin”, please change to “catechin”

Fig. 3. Is it the effect of antioxidants or of Vitamin C?

Line 205: “Anoectochilus formosanus”, please in italics

Lines 252/253: “in order to function”, in order to enable Trx to function, please make it clear

Line 760: “contribute”, please change to “contributes”

Line 835: “;Chaetomium”, please in italics

The language requires check and minor amendments

Author Response

(The authors gave the same response as above.)
